# Quality of Life in Women after Deep Endometriosis Surgery: Comparison with Spanish Standardized Values

**DOI:** 10.3390/jcm11206192

**Published:** 2022-10-20

**Authors:** Alicia Hernández, Elena Muñoz, David Ramiro-Cortijo, Emanuela Spagnolo, Ana Lopez, Angela Sanz, Cristina Redondo, Patricia Salas, Ignacio Cristobal

**Affiliations:** 1Department of Obstetrics and Gynecology, Hospital Universitario La Paz, Paseo de la Castellana, 261, 28046 Madrid, Spain; 2Research Institute of Hospital Universitario La Paz (IdiPaz), C/ Pedro Rico 6, 28019 Madrid, Spain; 3Department of Obstetrics and Gynecology, Hospital Universitario Puerta de Hierro, C/ Joaquín Rodrigo 1, Majadahonda, 28222 Madrid, Spain; 4Department of Physiology, Faculty of Medicine, Universidad Autónoma de Madrid, C/ Arzobispo Morcillo 2, 28029 Madrid, Spain

**Keywords:** deep endometriosis, laparoscopy, quality of life, surgery, short form of 12 items

## Abstract

The quality of life (QoL) of women who have been surgically treated for endometriosis may be severely impaired. Therefore, QoL can be a determining factor in the recovery of these patients. The aims of this study were to evaluate if the QoL of women surgically treated for deep endometriosis differs from a healthy age-matched population from Catalonia (Spain) and to analyze the QoL of these women considering concomitant events. This is an observational cross-sectional study, where 112 women (between 18 and 48 years old), with endometriosis treated by surgery at Hospital Universitario La Paz (Madrid, Spain), were enrolled to assess the QoL using the second version of the 12-item short form (SF-12) questionnaire. The QoL in these women were tested against a reference population of healthy women using a standardized one-sample comparison method. In addition, the QoL was compared according to the pathophysiology and type of surgery. In women with endometriosis, the physical health component, but not mental health component, was positively correlated with age (r = 0.19; *p*-Value = 0.048). In addition, physical (20.3 ± 29.2) and social functions (29.7 ± 38.3) and the overall physical health component (37.8 ± 19.4) were significantly lower than the reference population. On the contrary, the body pain (64.1 ± 41.2), emotional role (62.5 ± 42.2), mental health (54.4 ± 26.0), vitality (59.3 ± 31.2), and the overall mental health component (59.4 ± 26.6) had significantly higher scores than the reference. The anatomical compartment of endometriosis, reintervention, bowel nodule resection, and fertility preservation did not show statistical differences in QoL. Women with deep endometriosis had worse physical and social functions, and the overall physical health, compared to the norm in Spanish women. Bodily pain, emotional role, vitality, and the overall mental health improved. These areas could be considered protective factors in this disease. Considering the importance of QoL in adjustments in mental and physical health, it would be necessary to improve these areas of QoL in women surgically treated for deep endometriosis.

## 1. Introduction

Endometriosis is one of the most frequent chronic and benign gynecological diseases that affects childbearing-age women [1]. In this disease, the tissue from the endometrium is implanted outside the uterus, causing adhesions, fibrosis, and lesions in organs, such as the intestine or bladder [2]. Deep endometriosis is defined as the presence of endometrial tissue that infiltrates more than 5 mm beneath the peritoneum [3,4], affecting between 3–37% of premenopausal women with endometriosis [3,5,6,7]. Classic symptoms of endometriosis are pelvic pain and dysmenorrhea [2]. Menstrual disorders occur in 20–25% of the women with endometriosis, and sterility and infertility affect 30% of these women. In addition, anxiety or depression disorders occur in 60% of these patients [2].

Pharmacological or surgical treatments are available, but various factors have to be taken into account, such as the severity of the symptoms, age, degree and location of the lesions, previous response to medications, and the patient’s reproductive desire [8]. Pharmacological treatment is often the first choice, being frequently ineffective as deep endometriosis implants are made up of a large fibrous component [8,9]. Surgery is used in advanced cases, such as those that produce intestinal obstruction or ureteral stenosis, in progressive growth of the lesions, or in patients with little response to previous treatments [2]. In the short term, the quality of life in women with endometriosis treated surgically and pharmacologically improves [10]. However, in the long term, surgery is associated with severe postoperative events [11,12] that could diminish a woman’s quality of life. These events can influence women’s overall health by worsening their physical and mental components. To date, there are no studies that compare the QoL of these women after surgery with the standards in a healthy population.

Quality of life (QoL) is defined as the perception of each individual to assign value to the duration of life influenced by diseases, deficiencies, injuries, or treatments [13]. QoL is more and more relevant not only to determine the physical, social, and mental health of patients, but also to analyze the clinical procedures performed. Chronic and recurrent symptoms of endometriosis patients impair their ability to work, influencing family and social life [2,14]. Therefore, surgical treatment can be an option for women whose deep endometriosis symptoms reduce their QoL and are not controlled by medical treatment [15,16].

The QoL measurement focuses on assessing the degree of personal satisfaction in physical and social functions, physical and emotional roles, bodily pain, vitality, and mental and general health. Several scales have been designed, among the most used being the 36-item short form (SF-36) questionnaire or its reduced version SF-12, which is validated for the Spanish population [17,18].

The aim of this work was to study if there are differences in the QoL of women with deep endometriosis who were treated surgically with respect to an age-matched population established in Catalonia. Secondarily, this work explores whether the pathophysiological characteristics of deep endometriosis and the type of surgery determine differences in post-surgical QoL.

## 2. Materials and Methods

### 2.1. Study Design and Cohort Enrollment

This is an observational and cross-sectional study, which was conducted in women diagnosed with any deep endometriosis. This study was performed in the endometriosis unit of Hospital Universitario La Paz (HULP, Madrid, Spain). A total of 160 women participating in follow-up at this unit from 2010 to 2017 were candidates for entry into the study.

A woman was phoned by a researcher (E.M.) to participate if she was between 18 and 50 years old, had been diagnosed with deep endometriosis using gynecologic exploration or transvaginal ultrasound, and underwent surgery with subsequent histological confirmation. By phone, the researcher read the patient information sheet and, if it was convenient, the women approved the informed consent. After this step, the women answered the SF-12 questionnaire regarding QoL (see below). The phone calls were performed from January to March 2018, and the interview lasted about 20 min., approximately.

The socio-demographic and medical data were available in the database of the HULP. If the medical records were not available, the observation was withdrawn from the study (n = 3). The medical records of the women who had not consented to participate were not included in this study (n = 1). Also, women who did not respond to calls on more than three occasions were dropped from recruitment (n = 44). The final cohort was 112 women (participation rate of 70.0%).

This study was approved by the Research Ethics Committee of HULP on 26 January, 2018 (PI-3030).

### 2.2. Sociodemographic and Clinical Variables

From the medical records were recorded: age (years), time from the surgical intervention to SF-12 responses (years), gravida, parity (nulliparous, yes/no), previous miscarriages (yes/no), concomitant events (including anxiety, major depressive disorder, fibromyalgia, myofascial syndrome, chronic pelvic pain, infertility, polycystic ovary syndrome, autoimmune diseases, fibroid uterus, Crohn’s disease, and HIV), surgical reintervention (yes/no), surgical approach (considering laparoscopy versus laparotomy), bowel nodule resection (categorized as no, discoid, segmental, and shaving), cystectomy (categorized as no, uni-, or bi-lateral), ureterolysis (yes/no), and diagnoses of comorbidities. In addition, the intraoperative and postoperative complications were recorded, and they were classified as infections (abscesses and peritonitis), rectovaginal fistulas, ureteral injuries, bladder atony, hemorrhages (wall hematoma and rectal bleeding), anastomosis dehiscence (sigma opening, vaginal vault dehiscence, leak anastomotic, and uterine perforation), and fever. 

#### 2.2.1. Type of Surgery 

Deep endometriosis infiltrating nodes are removed during surgery per hospital protocol. The women were categorized according to preservation of (1) the uterus (regardless of ovarian involvement, hysterectomy), (2) both ovaries (regardless of whether they preserve the uterus, double adnexectomy), and (3) fertility, maintaining both uterus and ovaries (not including patients for whom any of these organs could be preserved).

#### 2.2.2. Compartment Affectation 

The degree of extension of the endometriosis was classified according to the involvement of the compartments as: anterior (bladder, vesico-uterine plica), posterior (uterosacral ligaments, uterine torus, rectovaginal septum, rectum, sigmoid), and lateral (ureters, sciatic nerve, parametria, iliac vessels, and round ligaments).

### 2.3. Quality of Life by Short-Form 12 (SF-12)

The SF-12 questionnaire is a reduced version of the SF-36 health questionnaire [18]. This study used the second version, developed in 2002 and adapted to the Spanish population [19]. Considering the time of the evaluation for this scale (four weeks), a single variation was applied since, in some cases, the SF-12 was carried out years after surgery. The time required to complete the SF-12 questionnaire was approximately 2 min. Research has verified that SF-12 is a valid and reliable measure, finding internal consistency estimates above 0.70 and significant correlations between the different versions of the scale [20,21].

The SF-12 questionnaire evaluates eight dimensions: (1) physical function (2 items evaluating the perceived physical effort), (2) physical role (2 items exploring the degree to which physical health interferes with work activities), (3) body pain (1 item, evaluating current physical pain), (4) social function (1 item exploring social difficulties), (5) emotional role (2 items evaluating the emotional adjustment for daily activities), (6) mental health (2 items exploring the degree of sadness and discouragement), (7) vitality (1 item exploring the degree of overall energy), and (8) general health (1 item, a personal assessment of health). 

The SF-12 combines both dichotomous (0 = yes; 1 = no) and 3 or 5-pointsLikert responses. The scores were standardized using the following Equation (1). The highest score, the best state of health, makes it possible to represent a percentage of the values, where X_i_ is the score for women i, Min is a minimum score of the variable, and Max-Min is the range of the variable [22].
Standardized score = ((X_i_-Min)/(Max − Min)) × 100(1)

Additionally, two summative components can be obtained from these 8 dimensions: the overall physical and mental health components [23]. The components calculation used the standardized scores of the dimensions weighted by their factor loading of principal component analysis with orthogonal rotation (Equations (2) and (3)), following the recommendations of the original authors of the SF-36 [19]. This study did not use T-scores. In this study, the internal consistency of SF-12 by Cronbach’s alpha was 0.83.
SPC = 0.407 × PF + 0.359 × FR + 0.332 × BP + 0.292 × GH + 0.040 × VT + 0.031 × SF − 0.242 × MH − 0.240 × ER(2)
SMC = −0.219 × PF − 0.163 × FR − 0.133 × BP − 0.069 × GH + 0.232 × VT + 0.241 × SF + 0.536 × MH + 0.512 × ER(3)

Being: summative physical component (SPC); summative mental component (SMC); physical function (PF); physical role (PR); body pain (BP); general health (GH); vitality (VT); social function (SF); mental health (MH); and emotional role (ER).

The scores of QoL according to the SF-12 were compared based on a specific method, which was the mean and standard deviation of the Catalonian population of healthy women according to age [17]. The means for women between 35 to 44 years old in the different dimensions of SF-12 is shown in Table 1.

### 2.4. Statistical Analysis

The quantitative variables were expressed as mean and standard deviation (SD), and the qualitative variables were expressed as relative frequency (%) and sample size (n). The distribution of the variables was checked using Shapiro’s test. To achieve the objective of studying the differences in the QoL of women with deep endometriosis with respect to a reference value, a one-sample Student’s *t*-test was used considering the norms of the Catalan women population of a similar age [17]. The magnitude of the differences between the present study and the reference value provides information about the deviation in our cohort. It was calculated by the ratio between the standardized mean of present data and the references’ norm-based score, following previous effect calculation [24]. Small was considered <0.2, between 0.2 to 0.5 was considered moderate and, >0.5 was considered large. In addition, Pearson’s coefficient (r) was used to demonstrate correlations.

In addition, to explore if the pathophysiological characteristics and the type of surgery (hysterectomy, double adnexectomy, and fertility-preserving resection) could establish differences in the post-surgical QoL, an independent Student’s *t*-test was applied. In multi-categorical variables, a one-way ANOVA was carried out for the different dimensions of SF-12 and the clinical variables using Dunnett’s post-hoc test. A *p*-value less than 0.05 was considered statistically significant. 

All analyses have been performed using R software (version 4.1.1, R Core Team 2021, Vienna, Austria) within RStudio (Version 1.4.1717, RStudio, PBC, 2009–2021, Inc., Vienna, Austria) and using the *rio*, *rcompanion*, *dplyr*, *ggplot2*, *corrplot*, *psych*, and *ggpubr* packages.

## 3. Results

### 3.1. Demographic and Clinical Characteristics

Women aged 35.5 ± 6.0 (min = 20, max = 48) years all presented symptoms suggestive of endometriosis. Laparoscopy was used in 97.3% (109/112) of surgeries, with 2.7% (3/112) performed with laparotomy due to the difficult control of intraoperative bleeding. The mean of previous pregnancy was 0.6 ± 0.8 (min = 0, max = 3). In addition, 65.2% (70/112) of women were nulliparous, 34.8% (39/112) had a delivery, and 11.6% (13/112) had a miscarriage. The period between surgery and the SF-12 response had an average length of 3.1 ± 2.3 years.

Regarding surgical procedures, 46.4% (52/112) involved a hysterectomy, 8.9% (10/112) a unilateral adnexectomy, 30.4% (34/112) a bilateral adnexectomy, 4.5% (5/112) involved partial bladder resection, 4.5% (5/112) ureteral reimplantation, and 18.8% (21/112) included a partial colpectomy. As for women who presented bowel nodule (rectum, sigmoid, or recto-sigma), the segmental resection was the most frequent procedure (35.7%), followed by shaving (25.0%) and discoid resection (10.7%). Ileostomy was required in 9.8% of segmental intestinal resections. In addition, the rate of surgeries preserving the uterus and ovary was 52.7% (59/112). Additionally, ureterolysis was performed in 22.3% of patients, unilateral cystectomy in 17.9% of patients, and 17.0% of patients had a bilateral cystectomy. 

In regard to intra- and postoperative complications, 73.2% (82/112) did not show complications. However, 5.4% (6/112) showed infections, another 5.4% (6/112) showed ureteral injuries, 4.5% (5/112) showed wall hematomas and rectal bleeding, bladder atony and rectovaginal fistulae were shown in 3.6% (4/112) of the cohort, and 2.7% (3/112) of the women showed anastomosis dehiscence, with 1.8% (2/112) of the women having a fever. 

### 3.2. Quality of Life in Women with Deep Endometriosis Compared to a Reference Population

All dimensions of SF-12 were correlated, exceptionally physical and mental components, and physical components with emotional role and mental health dimensions (Appendix A). In addition, the physical health component was significant and positively correlated with the age of the women (r = 0.19, *p*-Value = 0.048), while the mental health component had no significant correlation with age (r = −0.10, *p*-Value = 0.291). The physical (r = 0.12) and mental health components (r = 0.05) were not significantly correlated with the time between surgery and SF-12 response (*p*-value = 0.212 and *p*-value = 0.637, respectively; see Appendix A).

In our cohort, physical function (20.3 ± 29.2) and social function (29.7 ± 38.3) were significantly lower scoring dimensions compared to the reference population. In addition, the physical health component (37.8 ± 19.4) also scored significantly lower than the reference population (Table 2). However, the dimensions of body pain (64.1 ± 41.2), emotional role (62.5 ± 42.2), mental health (54.4 ± 26.0), and vitality (59.3 ± 31.2) had significantly higher scores than the reference population. Thus, the mental health component (59.4 ± 26.6) also scored significantly higher in women of the present sample compared to the reference (Table 2). The deviation was moderate in physical function, and the magnitude of the deviation in the rest of the dimensions and components was large.

### 3.3. Type of Surgery and Quality of Live in Women with Deep Endometriosis

None of the dimensions or components analyzed by SF-12 showed significant differences between women who preserved the uterus or were hysterectomized, in women with preservation of both ovaries compared to women with a double adnexectomy, and in those who kept the uterus and ovaries to preserve fertility compared to those who removed these organs (Table 3). 

### 3.4. Secondary Clinical Outcome Related to Quality of Life in Women with Deep Endometriosis 

The anatomical area according to the affected compartment was 8.9% (10/112) anterior, 87.5% (98/112) posterior, and 82.1% (92/112) lateral. Women with anterior compartment involvement scored significantly lower in social function (10.0 ± 21.1) compared to women without anterior compartment involvement (31.6 ± 39.2, *p*-value = 0.013). Women with posterior compartment involvement did not show statistical difference in any dimension or components nor with women without this affected compartment. Furthermore, women with lateral compartment involvement scored significantly lower in emotional role (59.2 ± 43.2) compared to women without lateral compartment involvement (77.5 ± 34.3, *p*-value = 0.048). The remaining dimensions and components showed no significant differences between the groups (Table 4).

No statistically significant differences were detected in any of the dimensions or components of the SF-12 between women with and without concomitant events, nor between women with and without reintervention (Appendix A). 

The emotional role was significantly lower among women who underwent laparotomy intervention (50.0 ± 0.0) compared to those who underwent laparoscopy (62.8 ± 42.7, *p*-Value = 0.002). In addition, the vitality was significantly lower among women who underwent laparotomy intervention (33.3 ± 11.5) compared to those who underwent laparoscopy (60.0 ± 31.3, *p*-value = 0.038) (Appendix A). The mental health component was lower in women intervened with laparotomy (45.5 ± 2.05) than those intervened with laparoscopy (59.8 ± 26.9, *p*-value < 0.001; Table 5).

No significant differences were detected in any of the dimensions or components of the SF-12 between women with bowel nodule resection (Appendix A).

## 4. Discussion

Quality of life is a multi- and interdisciplinary fact. Additionally, QoL is subjective and depends on how a woman perceives their own life. The decrease in QoL has been associated with a deterioration of health and poor physical well-being [13,25,26]. In women with deep endometriosis, an adverse impact on physical, mental, and social well-being [27] has been reported, as well as a negative effect on QoL [28]. This study showed altered scores in physical and social functions, body pain, emotional role, and vitality. In addition, altered overall physical and mental components of QoL were detected compared to those in a reference cohort of healthy women. The magnitude effect was large in body pain, emotional role, and mental health. A systematic review concluded the annually needs for QoL assessment in women with endometriosis for a proper clinical follow-up [29]. For this reason, these results would be important in considering the expanded period of evaluating QoL since endometriosis surgery.

Over the years, there have been investigations of the QoL of women with deep endometriosis and how surgical and pharmacological interventions influence the areas of QoL. Abbot et al. showed an improvement in the physical and mental components 5 years after surgery, although without significantly differences in the baseline [30]. Vercellini’s pharmacological study could also not demonstrate significant longitudinally differences in these components [31]. Considering our data, which were reported an average of 3 years after surgery (physical = [34.2; 41.5], mental = [54.4; 64.4]); the Abbot study (physical = [38.2; 54.1], mental = [38.6; 55.4]) and Vercellini’s study (physical = [52.8; 55.8], mental = [47.2; 57.4]); it can be seen that our population scored lower in the physical health component, but higher in the mental health component. These differences could be attributed to the severity of symptoms between populations that obstruct the perception of physical health and coping processes that improve over time. 

Additionally, we compared the QoL obtained by the SF-12 with the reference values based on the Catalonian population in 2012 [17]. This strategy was used since there are no standardized scores for women with endometriosis. In addition, we did have previous scores of these women before surgery. Both the standardization with the reference population and the time elapsed since surgery allow us to understand the process of QoL in women diagnosed with the disease during subsequent events following surgery. It is important to note was the rate of participation obtained is considered high, considering a similar study achieved a 40% rate of participation [32]. Additionally, although the scores of the reference population were obtained in 2012, it has been demonstrated that these scores were stable over a 10-year period in patients surgically treated for lumbar spine disease [33]. However, we did not check the long-term stability of the QoL profiles of the women in our cohort, which would be interesting research. 

Endometriosis affects a high number of women of a reproductive age. Fertility preservation should be a clinical option in women that express their desire for pregnancy. However, endometriosis surgery does not increase the chance of a successful pregnancy [34]. According to our data, the women who was surgery intervened with fertility preservation, obtained better scores in body pain, emotional role, general health, and overall mental health. Even these results were not significant, and these data are positive due to the impact of QoL on women’s well-being and illness-related coping. Similar results were found by other authors [8,9].

In women with endometriosis, the impasse of ovarian preservation often arises. Factors such as age, desire for pregnancy, and symptoms must be considered. A women’s age plays an important role in the decision, since hormonal suppression in 48-year-old women (maximum age of this cohort) does not have the same impact on QoL and complications as in 20-year-old women (minimum age of this cohort). In general, it needs to be considered that the degree of the effect of health on QoL increases with age [35]. As is well known, early menopause increases the risk of mortality due to cardiovascular events, fractures, and colorectal cancer, and it is related to worse QoL [36]. According to our data, overall physical health was significant and positively correlated with age of the women. These data suggest that in women with deep endometriosis, the option of fertility-preserving surgery could be a good therapeutic alternative that even improves women’s QoL. However, this effect needs further research due to the overall mental health in terms of QoL did not show a correlation with the age of women. It is known that factors affecting the mental component of QoL are a higher educational level—particularly health literacy—self-efficacy, physical health-promoting behavior, perceived emotional support, and few associated comorbidities [37], which were non-controlled factors in this study.

In the past, several studies focused on fertility and the desire for pregnancy relative to ovarian preservation and not on uterine preservation [8,28]. In recent years, authors investigated the importance of uterine-sparing surgery to achieve a lasting improvement of QoL [38,39]. In our cohort, the analysis of QoL in different types of surgery represents a strength. Even without reaching statistical significance, it was observed that women surgically treated by hysterectomy got worse scores in body pain, emotional role, and mental and general health compared to women whose uterus was preserved. However, it appears that the conditioning variable for physical and mental health in hysterectomy would be repeat surgery [40]. Regarding women surgically treated with a double adnexectomy, we found worse body pain, vitality, and overall mental health, without being statistically significant. Considering these factors, it is important to perform endometriosis surgery at referral health care hospitals and to prefer ovarian cystectomies in young women. Furthermore, the improvement of QoL from preserving female reproductive organs should be a challenge for endometriosis surgery [41].

The different pelvic compartments’ involvement in endometriosis have been explored in relation to the areas of QoL. Other authors observed an improvement in all areas of QoL in women surgically treated for posterior compartment endometriosis [15,42,43]. This data is essential considering the disabling symptomatology when endometriosis involves the posterior compartment [44]. According to our data, mental health was close to being significantly improved in women affected in the posterior compartment. In addition, social function and emotional role were depressed in women with affected endometriosis in anterior (typically bladder involvement) and lateral compartments (typically uterine and in parametrium tissue, important for urination, defecation, and sexual function), respectively. This data could suggest that, after surgery, some nerves are damaged, and these women may face weeks of self-catheterization, vaginal dryness, and defecation disorders. However, it should be considered that, as with women in our cohort, women with posterior compartment involvement frequently also experience the involvement of the lateral compartment [45,46].

With respect to the surgery used to remove colorectal nodules, we did not find significant differences in QoL. In the literature, the association between the technique of nodule resection (shaving, discoid or segmental) and health and mental functions is not clear [42,43,47]. Interestingly, in a retrospective Wullschleger’s study performed on 211 women (Appendix A), although the technique of colorectal surgery was not mentioned, an increase in working performance was shown after endometriosis surgery [32]. This is important, as working performance has key consequences on the perception of emotional roles and mental, physical, and social functions [48]. A study of women with endometriosis treated by laparoscopic reported higher overall mental scores than their controls, near statistical significance [49]. According to our data, the laparoscopic surgical approach improves the emotional role, vitality, and overall mental health, although these data require further investigation. 

In clinical practice with women who suffer deep endometriosis, routine evaluation of QoL should be considered essential both for the health-care provider and the women themselves. Nevertheless, the limitations of the present study need to be considered. Firstly, based on the retrospective design, we did not have the possibility of carrying out the SF-12 questionnaire before surgery. However, the strategy of standardization and comparison with a population of healthy women could solve this issue, considering also that differences in the physical and mental health components of SF-12 have not been demonstrated longitudinally [30,31]. Secondly, the impossibility of comparing the scores obtained in our cohort with the standard in populations of women with deep endometriosis must be noted for the SF-12. 

In contrast, the time (around three years) between the surgery and the SF-12 responses can represent a strength of this study, because this period allowed the women to recover from surgery, to return to work and social life, to continue with hormonal treatment, or to become pregnant. All these factors are extremely important for the evaluation of women’s QoL.

## 5. Conclusions

This study sees that, compared to the norm in a Spanish population, women with deep endometriosis treated surgically had worse physical and social functions. Overall, the physical health component of the quality of life was depressed. However, as the woman’s age increases, the perception of improved physical health also increases. In these women, bodily pain, emotional role, mental health, and vitality were increased. Furthermore, the mental health component of quality of life also was increased. These areas could be considered protective factors in this disease prognosis, considering also that the type of surgery did not affect the patients’ quality of life. It would be necessary to attend to the quality of life of women with endometriosis as a key parameter to balance the appropriate treatment based on women’s needs

## Figures and Tables

**Table 1 jcm-11-06192-t001:** Catalonian population norms of the SF-12 dimensions for women aged 44–53 years in 2012.

	Mean (SD)	CI [95%]		Mean (SD)	CI [95%]
Physical function	51.7 (8.3)	[50.8; 52.5]	Emotional role	50.2 (9.1)	[49.3; 51.1]
Physical role	50.9 (9.0)	[50.0; 51.8]	Mental health	49.5 (9.4)	[48.5; 50.4]
Body pain	50.7 (9.6)	[49.7; 51.7]	Vitality	49.6 (9.9)	[48.5; 50.6]
Social function	50.9 (8.7)	[50.0; 51.8]	General health	50.8 (9.1)	[49.9; 51.8]
Physical component	51.6 (8.1)	[50.7; 52.4]	Mental component	49.3 (9.5)	[48.3; 50.3]

Data shown: mean, standard deviation (SD), and 95% of confidence interval [CI]. Data extracted from [17].

**Table 2 jcm-11-06192-t002:** Differences in dimensions and components of SF-12 of women with deep endometriosis treated by surgery with respect to a norm of Catalan women.

	Reference	Mean (SD)	CI [95%]	Magnitude	*p*-Value
Physical function	51.7 (8.3)	20.3 (29.2)	[14.9; 25.8]	0.4	<0.001
Physical role	50.9 (9.0)	54.5 (45.3)	[46.0; 62.9]	1.1	0.407
Body pain	50.7 (9.6)	64.1 (41.2)	[56.3; 71.8]	1.3	0.001
Social function	50.9 (8.7)	29.7 (38.3)	[22.5; 36.9]	0.6	<0.001
Emotional role	50.2 (9.1)	62.5 (42.2)	[54.6; 70.4]	1.2	0.003
Mental health	49.5 (9.4)	59.4 (26.0)	[54.5; 64.3]	1.2	<0.001
Vitality	49.6 (9.9)	59.3 (31.2)	[53.4; 65.1]	1.1	0.001
General health	50.8 (9.1)	50.8 (34.0)	[44.4; 57.1]	1.0	0.989
Physical component	51.6 (8.1)	37.8 (19.4)	[34.2; 41.5]	0.7	<0.001
Mental component	49.3 (9.5)	59.4 (26.6)	[54.4; 64.4]	1.2	<0.001

Data shown: mean, standard deviation (SD), and 95% of confidence interval [CI]. The *p*-value was extracted with one-sample Student’s *t*-test considering the reference column from [17]. The magnitude column was calculated as the ratio between the score of the present study and the reference value.

**Table 3 jcm-11-06192-t003:** Dimensions and components of SF-12 among women with or without hysterectomy, double adnexectomy, and fertility-preserving node resection.

	Hysterectomy			Double Adnexectomy			Fertility-Preserving Resection		
	No (n = 60)	Yes (n = 52)	*p*-Value	No (n = 72)	Yes (n = 40)	*p*-Value	No (n = 57)	Yes (n = 55)	*p*-Value
Physical function	18.3 (28.3)	22.6 (30.2)	0.445	17.7 (26.7)	25.0 (33.0)	0.236	23.7 (30.8)	16.8 (27.2)	0.213
Physical role	55.0 (44.8)	53.8 (46.3)	0.894	54.9 (45.3)	53.8 (45.8)	0.902	53.5 (46.2)	55.5 (44.8)	0.821
Body pain	67.1 (40.0)	60.6 (42.7)	0.409	67.0 (38.6)	58.8 (45.5)	0.335	59.6 (43.2)	68.6 (39.2)	0.250
Social function	30.0 (37.6)	29.3 (39.5)	0.927	28.5 (36.4)	31.9 (42.0)	0.668	30.3 (39.2)	29.1 (37.8)	0.872
Emotional role	64.2 (40.2)	60.6 (44.7)	0.658	63.2 (41.1)	61.2 (44.6)	0.821	59.6 (22.8)	65.5 (39.5)	0.468
Mental health	61.1 (24.1)	57.5 (28.2)	0.469	60.3 (23.1)	57.8 (30.9)	0.649	56.3 (29.0)	62.6 (22.4)	0.201
Vitality	59.3 (30.1)	59.2 (32.7)	0.986	60.3 (29.7)	57.5 (33.9)	0.666	60.0 (33.6)	58.5 (28.8)	0.806
General health	53.5 (32.4)	47.6 (35.9)	0.366	51.4 (33.1)	49.6 (36.1)	0.799	48.2 (36.0)	53.4 (32.0)	0.428
Physical component	38.2 (20.4)	37.4 (18.4)	0.818	37.7 (19.1)	38.1 (20.1)	0.921	38.1 (19.2)	37.5 (19.8)	0.865
Mental component	61.0 (26.1)	57.6 (27.4)	0.499	60.3 (23.9)	57.9 (31.3)	0.676	56.8 (29.0)	62.1 (23.9)	0.287

Data shown: mean and standard deviation (SD). The *p*-value was extracted by independent Student’s *t* test.

**Table 4 jcm-11-06192-t004:** Dimensions and components of SF-12 among women with affected compartment of deep endometriosis.

	Anterior			Posterior			Lateral		
	No (n = 102)	Yes (n = 10)	*p*-Value	No (n = 14)	Yes (n = 98)	*p*-Value	No (n = 20)	Yes (n = 92)	*p*-Value
Physical function	20.6 (29.1)	17.5 (31.3)	0.770	32.1 (35.9)	18.6 (27.9)	0.196	20.0 (34.0)	20.4 (28.2)	0.963
Physical role	53.9 (45.4)	60.0 (45.9)	0.697	39.3 (44.6)	56.6 (45.2)	0.192	62.5 (45.5)	52.7 (45.3)	0.391
Body pain	63.0 (41.3)	75.0 (40.8)	0.394	50.0 (49.0)	66.1 (39.9)	0.259	73.8 (41.7)	62.0 (41.0)	0.260
Social function	31.6 (39.2)	10.0 (21.1)	0.013	41.1 (41.1)	28.1 (37.9)	0.280	25.0 (40.6)	30.7 (38.0)	0.569
Emotional role	61.8 (42.3)	70.0 (42.2)	0.568	60.7 (40.1)	62.8 (42.7)	0.862	77.5 (34.3)	59.2 (43.2)	0.048
Mental health	59.8 (26.0)	55.6 (27.2)	0.646	48.4 (24.5)	61.0 (26.0)	0.092	65.0 (25.8)	58.2 (26.1)	0.296
Vitality	60.2 (31.1)	50.0 (31.6)	0.351	50.0 (37.4)	52.9 (33.1)	0.326	59.0 (30.8)	59.3 (31.4)	0.964
General health	51.3 (33.9)	45.0 (36.5)	0.610	36.1 (37.9)	52.9 (33.1)	0.136	53.8 (34.9)	50.1 (34.0)	0.674
Physical component	37.7 (19.5)	38.8 (19.5)	0.875	31.3 (24.9)	38.8 (18.5)	0.298	39.6 (17.3)	37.4 (19.9)	0.633
Mental component	60.0 (26.6)	52.9 (28.2)	0.461	56.0 (29.6)	59.9 (26.3)	0.642	66.1 (22.3)	57.9 (27.4)	0.163

Data shown: mean and standard deviation (SD). The *p*-value was extracted with an independent Student’s *t*-test.

**Table 5 jcm-11-06192-t005:** Dimensions and components of SF-12 among women with concomitant events, reintervention, and surgical approach.

	Concomitant Events			Reintervention			Surgical Approach		
	No (n = 83)	Yes (n = 28)	*p*-Value	No (n = 53)	Yes (n = 58)	*p*-Value	LPS (n = 109)	LPT (n = 3)	*p*-Value
Physical function	21.1 (30.6)	18.8 (25.1)	0.690	17.5 (26.7)	23.3 (31.4)	0.293	20.4 (29.3)	16.7 (28.9)	0.844
Physical role	56.0 (45.8)	48.2 (44.1)	0.426	55.7 (43.5)	52.6 (47.2)	0.722	55.0 (45.1)	33.3 (57.7)	0.583
Body pain	65.1 (41.9)	59.8 (39.9)	0.556	66.5 (37.3)	61.2 (44.7)	0.498	64.7 (41.0)	41.7 (52.0)	0.525
Social function	26.8 (37.4)	39.3 (40.5)	0.158	26.5 (36.3)	32.8 (40.3)	0.421	29.6 (38.1)	33.3 (57.7)	0.921
Emotional role	63.3 (42.8)	58.9 (40.9)	0.635	63.2 (40.6)	61.2 (44.0)	0.804	62.8 (42.7)	50.0 (0.0)	0.002
Mental health	58.8 (26.8)	59.9 (23.3)	0.829	59.5 (23.5)	58.6 (28.1)	0.852	59.9 (25.8)	40.7 (33.9)	0.431
Vitality	60.5 (31.5)	55.0 (30.6)	0.420	62.3 (28.7)	56.2 (33.4)	0.307	60.0 (31.3)	33.3 (11.5)	0.038
General health	52.5 (33.5)	43.8 (34.7)	0.249	50.7 (32.9)	50.0 (35.0)	0.919	50.9 (34.2)	45.0 (34.6)	0.797
Physical component	38.5 (19.3)	32.4 (19.3)	0.097	37.7 (18.3)	37.7 (20.6)	0.990	38.1 (19.4)	26.2 (18.2)	0.374
Mental component	58.4 (27.9)	61.6 (22.8)	0.544	60.0 (24.4)	58.4 (28.7)	0.762	59.8 (26.9)	45.5 (2.04)	<0.001

Data shown: mean and standard deviation (SD). The *p*-value was extracted with an independent Student’s *t*-test. Laparoscopy (LPS), Laparotomy (LPT).

## Data Availability

The data presented in this study are available on request from the corresponding authors. The availability of the data is restricted to investigators based in academic institutions.

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
