# Peer review of "Quality of Life in Women after Deep Endometriosis Surgery: Comparison with Spanish Standardized Values"

_jcm, 2022, doi:10.3390/jcm11206192_

Round 1

Reviewer 1 Report

1-The manuscript should be thoroughly proofed for language and grammar.
2-Unexplained acronyms should not be used in the Abstract, Highlights, or as Keywords, and all acronyms should be defined in the body text.                                                                          3-More recent studies about endometrioma can be referred to. Suggested;

-https://doi.org/10.1016/j.saa.2021.120246

-https://doi.org/10.1016/j.saa.2022.121119 

4-Table legends should be more explanatory and also table organizations are messy, it should be arranged.

Author Response

  1. The manuscript should be thoroughly proofed for language and grammar.

Response: Thank you for your time reviewing our work. The article has been extensively reviewed.

  1. Unexplained acronyms should not be used in the Abstract, Highlights, or as Keywords, and all acronyms should be defined in the body text.

Response: The acronyms in the abstract and keywords have been replaced by their definitions.                           

  1. More recent studies about endometrioma can be referred to. Suggested.
    • https://doi.org/10.1016/j.saa.2021.120246
    • https://doi.org/10.1016/j.saa.2022.121119

Response: Thank you for these valuable articles. However, they are focus on the diagnosis of endometriosis through PCA techniques and lipidomic analysis. Our study does not assess the diagnosis but the quality of life of women who were diagnosed with deep endometriosis.

  1. Table legends should be more explanatory and also table organizations are messy, it should be arranged.

Response: The tables and legends have been modified and edited.

Reviewer 2 Report

This manuscript discusses postoperative quality of life in cases of deep endometriosis. While this study is of interest, the objectives are not clear and it is unclear what the results of this study are intended to reveal. So it is unclear to the reader what this conclusion is trying to say. If the objectives of this study are revised and clarified, the conclusions will naturally coalesce. It is also unclear why the target cases were focused on deep endometriosis. In addition, it does not state whether the procedure involved removal of deep endometriosis or not. It is not at all clear whether this study shows the effect of surgery on normal endometriosis or reveals the effect of having deep endometriosis removed. Also, once the purpose is clearly defined, data not needed for that purpose should be omitted. This manuscript is not a "deep endometriosis" review. Since the authors seem to have preoperative data, a comparison between preoperative and postoperative must have yielded better results.

This manuscript needs to be revised in a fundamental way.

The following is a list of points to be corrected.

Abstract

The objectives of this study were to compare the QoL of women with endometriosis ~ and to assess this QoL of women considering the concomitant events. The aims of this study were to compare the QoL of women with endometriosis ~ and to assess this QoL of women considering the concomitant events. The purpose is what you want to do by comparing them. It should be stated specifically, not "to assess this QoL. The conclusion is completely useless: "It is necessary to attend to the quality of life in the follow-up of women with deep endometriosis. It should be corrected.

Introduction

Facts that have been revealed so far are listed in detail, but there is no description of unknowns that have not been revealed so far. This is why the purpose of this study (what the authors are trying to clarify in this paper) is not clear.

Materials and Methods

Too many items have been extracted because it is not clear what this study will reveal. Items that are not relevant to the objective should be excluded. Three types of surgeries are classified, but it is unclear whether deep endometriosis is removed or not. It should be clarified.

Discussion

Similarly, the discussion of endometriosis and the discussion of deep endometriosis are mixed, and the results obtained from this paper are not utilized. If this paper were to revise its purpose, the discussion would naturally become more coherent.

Author Response

This manuscript discusses postoperative quality of life in cases of deep endometriosis. While this study is of interest, the objectives are not clear and it is unclear what the results of this study are intended to reveal. So it is unclear to the reader what this conclusion is trying to say. If the objectives of this study are revised and clarified, the conclusions will naturally coalesce. It is also unclear why the target cases were focused on deep endometriosis. In addition, it does not state whether the procedure involved removal of deep endometriosis or not. It is not at all clear whether this study shows the effect of surgery on normal endometriosis or reveals the effect of having deep endometriosis removed. Also, once the purpose is clearly defined, data not needed for that purpose should be omitted. This manuscript is not a "deep endometriosis" review. Since the authors seem to have preoperative data, a comparison between preoperative and postoperative must have yielded better results.

Response: Thank you for the time spent reviewing this work. Your suggestions have been discussed and answered. The objectives and the proposal of this work have been revised. However, we do not have preoperative data, since the quality of life questionnaire was done after surgery in all cases. The text has modified, and we believe it has improved.

This manuscript needs to be revised in a fundamental way. The following is a list of points to be corrected.

Abstract. The objectives of this study were to compare the QoL of women with endometriosis ~ and to assess this QoL of women considering the concomitant events. The purpose is what you want to do by comparing them. It should be stated specifically, not "to assess this QoL. The conclusion is completely useless: "It is necessary to attend to the quality of life in the follow-up of women with deep endometriosis. It should be corrected.

Response: The objective and conclusions have been edited to make the text more attractive and useful.

Introduction. Facts that have been revealed so far are listed in detail, but there is no description of unknowns that have not been revealed so far. This is why the purpose of this study (what the authors are trying to clarify in this paper) is not clear.

Response: Thank you for this guidance. Considering to that, in the short term, the quality of life in women with endometriosis treated surgically improves and pharmacological treatments lead to a decrease in endometriotic nodules avoiding recurrence of pain (PMID: 21131296), it remains unresolved whether, in the long term, these women achieve a quality of life similar to that of the population of healthy women, avoiding postoperative digestive and functional complications. We have modified the text to clarify the work proposal (lines 58-64).

Materials and Methods. Too many items have been extracted because it is not clear what this study will reveal. Items that are not relevant to the objective should be excluded. Three types of surgeries are classified, but it is unclear whether deep endometriosis is removed or not. It should be clarified.

Response: Deep endometriosis infiltrating nodes are always removed during surgery per clinical hospital protocol and guidelines. In this exploratory study we wanted to test a large number of variables to see on QoL both surgeries and postoperative adverse events might have effect.

Discussion. Similarly, the discussion of endometriosis and the discussion of deep endometriosis are mixed, and the results obtained from this paper are not utilized. If this paper were to revise its purpose, the discussion would naturally become more coherent.

Response: The discussion has been deeply revised and paragraphs referring to the original data reported in this paper have been integrated throughout the discussion.

Reviewer 3 Report

The study aimed to investigate, by means of SF-12 questionnaire, the quality of life (QoL), both under the physical and social function, of women affected by endometriosis who underwent surgery, in a cohort of Spanish patients. The study is interesting, since permitted to evaluate the QoL of women surgically treated for endometriosis in comparison to the characteristics of an age-matched population of Catalonia, as well as the role of concomitant events. Results are convincing and are presented and discussed clearly. However, I suggest some points, hereafter listed, that may be evaluated more in detail, especially under a methodological point of view.

Methods, Cohort enrollment: Was any power analysis performed, in order to check the appropriate sample size?

Line 79: The pool of candidates for entry into the study is reported among patients on follow-up in the period 2010-2017. However, except for the Ethics Committee approval date (January 2018), no information is given when the study was done (phone interview) and how long the study lasted. Only at line 181 and line 364 a mention to time between surgery and SF-12 evaluation (average of 3.1±2.3 years , “around three years”) is given.

Table 1: “Media” should be “Mean”; same at line 153. Also in Table 2, line 215, correct term is “mean”.

Table 1: The data from reference [14] are from 2012 survey; are there demonstrations that a 10-year period (considering reference at today, year 2022) does not affect the reference population data of SF-12? In the literature there seem to be reports of stability of the scores over short (2-3 years) time. Do the Authors have any data regarding longer time-stability issue of SF-12 in a given population? This point could be addressed in Discussion, lines 287-288.

Line 163: The range for considering the magnitude “large” for an effect, as indicated by deGraaf et al. [22] (page 2683) should be >0.8, referring to calculated effect sizes (Cohen’s d). An explanation for considering here the value >0.5 should be given, and if this 0.5 value is Cohen’s d or an absolute difference in magnitude.

Line 163: “Pearson’s coefficients (rho) was used to demonstrate correlations”. Usually, the sample statistic for Pearson’s coefficient is stated by the Roman letter r, while the population parameter is indicated by the Greek letter ρ (rho); rho is used also for Spearman rank correlation. To this end, did the Authors consider the use also of Spearman rank correlation rho to evaluate links between variables?

Lines 165-170: Did the Authors check whether the variables were normally distributed (for instance using Shapiro-Wilk test)? In case the assumption of normality is not accepted, comparison of means would benefit by using a non-parametric test (e.g. Kruskal-Wallis test, followed by post-hoc Steel’s test or Mann-Whitney test).

Line 176: Women were aged 35.5±6.0

Line 198: “all dimensions and components of SF-12 were correlated (data not shown).” A table reporting the correlations would be useful, for instance as supplementary material (see for instance table 3 in Huo et al., doi: 10.1186/s12955-018-0858-2).  Any reference to Cronbach's alpha reliability coefficient for internal consistency of test?

Line 216-217: The “magnitude” meaning is not clear.  Refer also to line 163 definition. Is this a difference between the raw (not standardized) score from present study and that from reference [14], or instead is a Cohen’s d? As it is reported, the meaning of an absolute “magnitude” is difficult to perceive.

Lines 229-230 and table 4: The number of patients with posterior or lateral anatomical involvement is 98+92=190, indicating that within the total 112 cases several patients had both compartments affected. I wonder if the comparisons no/yes presented in Table 4 could be biased, if a patient has both posterior and lateral involvement, so belonging to both groups. Moreover, did the Authors consider the issue of multiple comparisons, that suggests when more inferences are made, the more likely erroneous inferences occur; in particular, regarding the “Social function” item, which is the only one resulting as significant.

Table 5: The sample size in the “Surgical approach” is quite unequal between the two cases LPS/LPT, in particular due to the very low number (n=3) for LPT. I wonder if there is adequate statistical power in this situation. Maybe a non-parametric test would be another option. In particular, "Emotional role" in women who underwent laparotomy is reported as 50.0±0.0: was SD zero?

Line 278: A discussion is done comparing the present results with those of Abbot et al., comparing QoL after 5 years from surgery. The time from surgery for the present study should be remembered here, around line 281, in order to make a better comparison.

Lines 314-315: “overall mental health of QoL did not shown correlation with women age.” A correlation might be still present analysing data by using Spearman correlation. A preliminary scatter plot (it could be a good idea to present it as supplementary material) will suggest if there is a monotonic relationship or a non-linear relationship; the latter could be anyway of interest, although not well described by Pearson’s linear correlation coefficient.

Author Response

The study aimed to investigate, by means of SF-12 questionnaire, the quality of life (QoL), both under the physical and social function, of women affected by endometriosis who underwent surgery, in a cohort of Spanish patients. The study is interesting, since permitted to evaluate the QoL of women surgically treated for endometriosis in comparison to the characteristics of an age-matched population of Catalonia, as well as the role of concomitant events. Results are convincing and are presented and discussed clearly. However, I suggest some points, hereafter listed, that may be evaluated more in detail, especially under a methodological point of view.

Response: Thank you for the time spent reviewing this work. Your suggestions have been discussed and answered. The text has modified, and we believe it has improved.

Methods, Cohort enrollment: Was any power analysis performed, in order to check the appropriate sample size?

Response: this is a great comment. No sample size calculation was made because we wanted to obtain exploratory data on how women with endometriosis scored in their quality of life and we included all possible patients seen in the endometriosis unit of the hospital.

Line 79: The pool of candidates for entry into the study is reported among patients on follow-up in the period 2010-2017. However, except for the Ethics Committee approval date (January 2018), no information is given when the study was done (phone interview) and how long the study lasted. Only at line 181 and line 364 a mention to time between surgery and SF-12 evaluation (average of 3.1±2.3 years , “around three years”) is given.

Response: Thank you for this suggestion. The phone calls were made from January to March 2018. The telephone interview lasted about 20 min, approximately. This information has been updated in the manuscript (lines 93-94).

Table 1: “Media” should be “Mean”; same at line 153. Also in Table 2, line 215, correct term is “mean”.

Response: Thank you for your comment, the text has been edited.

Table 1: The data from reference [14] are from 2012 survey; are there demonstrations that a 10-year period (considering reference at today, year 2022) does not affect the reference population data of SF-12? In the literature there seem to be reports of stability of the scores over short (2-3 years) time. Do the Authors have any data regarding longer time-stability issue of SF-12 in a given population? This point could be addressed in Discussion, lines 287-288.

Response: This is a great point, there are studies showing that health questionnaire scores of QoL, such as SF-36 or short-version SF-12, are stable over a 10-year period. These scores were demonstrated in patients surgically treated for lumbar spine diseases (DOI: 10.1186/s12955-022-01999-7). However, we did not check the long-term stability of our QoL profiles, which it would be a great new research project. in This information was updated in the manuscript (lines 312-316).

Line 163: The range for considering the magnitude “large” for an effect, as indicated by deGraaf et al. [22] (page 2683) should be >0.8, referring to calculated effect sizes (Cohen’s d). An explanation for considering here the value >0.5 should be given, and if this 0.5 value is Cohen’s d or an absolute difference in magnitude.

Response: Thank you for this consideration. The effect size was calculated following the de Graaf´s study. However, it is not an effect size as described Cohen. Therefore, we considered a large effect size if it exceeded a ratio dispersion higher than 0.5. The text has been modified for better understanding (lines 170-174).

Line 163: “Pearson’s coefficients (rho) was used to demonstrate correlations”. Usually, the sample statistic for Pearson’s coefficient is stated by the Roman letter r, while the population parameter is indicated by the Greek letter ρ (rho); rho is used also for Spearman rank correlation. To this end, did the Authors consider the use also of Spearman rank correlation rho to evaluate links between variables?

Response: Thank you for this appreciation, “rho” was changed to “r” to indicate Pearson's correlations. On the other hand, Spearman´s correlations were not considered because the data distribution was corroborated by Shapiro test, and normality assumption was achieved.

Lines 165-170: Did the Authors check whether the variables were normally distributed (for instance using Shapiro-Wilk test)? In case the assumption of normality is not accepted, comparison of means would benefit by using a non-parametric test (e.g. Kruskal-Wallis test, followed by post-hoc Steel’s test or Mann-Whitney test).

Response: Thank you for this consideration. The variables were tested using the Shapiro test for normality. This was modified in the material and methods.

Line 176: Women were aged 35.5±6.0

Response: Thanks for the correction. The text was edited.

Line 198: “all dimensions and components of SF-12 were correlated (data not shown).” A table reporting the correlations would be useful, for instance as supplementary material (see for instance table 3 in Huo et al., doi: 10.1186/s12955-018-0858-2).  Any reference to Cronbach's alpha reliability coefficient for internal consistency of test?

Response: Thanks for this suggestion, we have added a supplementary table (table S1) with correlations between SF-12 scores and reliability coefficient in material and methods (line 151).

Line 216-217: The “magnitude” meaning is not clear.  Refer also to line 163 definition. Is this a difference between the raw (not standardized) score from present study and that from reference [14], or instead is a Cohen’s d? As it is reported, the meaning of an absolute “magnitude” is difficult to perceive.

Response: The text has been edited to correct the magnitude index.

Lines 229-230 and table 4: The number of patients with posterior or lateral anatomical involvement is 98+92=190, indicating that within the total 112 cases several patients had both compartments affected. I wonder if the comparisons no/yes presented in Table 4 could be biased, if a patient has both posterior and lateral involvement, so belonging to both groups. Moreover, did the Authors consider the issue of multiple comparisons, that suggests when more inferences are made, the more likely erroneous inferences occur; in particular, regarding the “Social function” item, which is the only one resulting as significant.

Response: This is a great observation. In women whose endometriosis affects the posterior compartment, usually it is found involvement of the lateral compartment (PMID: 21233128). It is recognized that the posterior nodule tends to infiltrate the lateral compartment (PMID: 28969477). Apart from the common occurrence of these compartments being simultaneously affected, what is essential is the disabling symptomatology related to endometriosis of the posterior compartment (PMID: 18971131). Therefore, the quality of life of women with posterior compartment involvement should be considered together with the lateral compartment. However, it is a very interesting comment which was included in the discussion (lines 358-360 and 366-368).

Table 5: The sample size in the “Surgical approach” is quite unequal between the two cases LPS/LPT, in particular due to the very low number (n=3) for LPT. I wonder if there is adequate statistical power in this situation. Maybe a non-parametric test would be another option. In particular, "Emotional role" in women who underwent laparotomy is reported as 50.0±0.0: was SD zero?

Response: He is right that the sample size was small in this comparison. However, all the work was performed with a parametric analysis of the results since the normality of the variables was checked. We should also consider that this analysis is part of a secondary outcome. Even so, let us give the analysis of the data using a nonparametric analysis.

Surgical approach

LPS (n=109)

LPT (n=3)

p-Value

Physical function

0.00 [0.00; 25.0]

0.00 [0.00; 25.0]  

0.816

Physical role

50.0 [0.00; 100] 

0.00 [0.00; 50.0]  

0.429

Body pain

100 [25.0; 100] 

25.0 [12.5; 62.5]  

0.399

Social function

0.00 [0.00; 75.0]

0.00 [0.00; 50.0]  

0.992

Emotional role

100 [0.00; 100] 

50.0 [50.0; 50.0]  

0.408

Mental health

66.7 [44.4; 77.8]

33.3 [22.2; 55.6]  

0.268

Vitality

60.0 [40.0; 80.0]

40.0 [30.0; 40.0]  

0.129

General health

60.0 [25.0; 85.0]

25.0 [25.0; 55.0]  

0.803

Physical component

41.9 [23.3; 53.5]

16.9 [15.8; 32.0]  

0.231

Mental component

67.1 [41.6; 78.5]

45.2 [44.4; 46.4]  

0.168

On the other hand, we rechecked the data and the SD of "emotional role" was 0.00.

Line 278: A discussion is done comparing the present results with those of Abbot et al., comparing QoL after 5 years from surgery. The time from surgery for the present study should be remembered here, around line 281, in order to make a better comparison.

Response: Great contribution, the text has been modified to remind the reader of this fact (line 298).

Lines 314-315: “overall mental health of QoL did not shown correlation with women age.” A correlation might be still present analyzing data by using Spearman correlation. A preliminary scatter plot (it could be a good idea to present it as supplementary material) will suggest if there is a monotonic relationship or a non-linear relationship; the latter could be anyway of interest, although not well described by Pearson’s linear correlation coefficient.

Response: Thanks for this approach, we have reported the scatter plot in the supplementary figure 1 along with the significance.

Round 2

Reviewer 2 Report

The first manuscript has been revised almost appropriately and this revised version is likely to be acceptable. However, several small modifications are considered necessary.

Those modifications are listed below.

1.       Abstract

Page 1, lines 19-20: "The aims of this study were to evaluate if the QoL of women with endometriosis surgically treated differs from~"

The term "deep endometriosis" should be used instead of "endometriosis.

2.       Materials and Methods

Page 3, lines 116-119: Type of surgery.⇒In this paragraph, the sentence " Deep endometriosis infiltrating nodes are always removed during surgery per clinical hospital protocol and guidelines." should be added.

Author Response

The first manuscript has been revised almost appropriately and this revised version is likely to be acceptable. However, several small modifications are considered necessary. Those modifications are listed below.

  1. Page 1, lines 19-20: "The aims of this study were to evaluate if the QoL of women with endometriosis surgically treated differs from~". The term "deep endometriosis" should be used instead of "endometriosis.
  2. Materials and Methods. Page 3, lines 116-119: Type of surgery. In this paragraph, the sentence "Deep endometriosis infiltrating nodes are always removed during surgery per clinical hospital protocol and guidelines." should be added.

Response: Thank you for your time reviewing our article. Your recommendations have been included in the revised version.